# Tourism Service Scheduling in Smart City Based on Hybrid Genetic Algorithm Simulated Annealing Algorithm

Pannee Suanpang [1,*], Pitchaya Jamjuntr [2], Kittisak Jermsittiparsert [3,4,5,6,7] and Phuripoj Kaewyong [1]

1   Faculty of Science & Technology, Suan Dusit University, Bangkok 10300, Thailand
2   Faculty of Engineering, King Mongkut's University of Technology Thonburi, Bangkok 10140, Thailand
3   Faculty of Education, University of City Island, Famagusta 9945, Cyprus
4   Faculty of Social and Political Sciences, Universitas Muhammadiyah Sinjai, Kabupaten Sinjai 92615, Sulawesi Selatan, Indonesia
5   Faculty of Social and Political Sciences, Universitas Muhammadiyah Makassar, Kota Makassar 90221, Sulawesi Selatan, Indonesia
6   Publication Research Institute and Community Service, Universitas Muhammadiyah Sidenreng Rappang, Sidenreng Rappang Regency 91651, South Sulawesi, Indonesia
7   Sekolah Tinggi Ilmu Administrasi Abdul Haris, Kota Makassar 90000, Sulawesi Selatan, Indonesia
*   Correspondence: pannee_sua@dusit.ac.th

**Abstract:** The disruptions in this era have caused a leap forward in information technology being applied in organizations to create a competitive advantage. In particular, we see this in tourism services, as they provide the best solution and prompt responses to create value in experiences and enhance the sustainability of tourism. Since scheduling is required in tourism service applications, it is regarded as a crucial topic in production management and combinatorial optimization. Since workshop scheduling difficulties are regarded as extremely difficult and complex, efforts to discover optimal or near-ideal solutions are vital. The aim of this study was to develop a hybrid genetic algorithm by combining a genetic algorithm and a simulated annealing algorithm with a gradient search method to the optimize complex processes involved in solving tourism service problems, as well as to compare the traditional genetic algorithms employed in smart city case studies in Thailand. A hybrid genetic algorithm was developed, and the results could assist in solving scheduling issues related to the sustainability of the tourism industry with the goal of lowering production requirements. An operation-based representation was employed to create workable schedules that can more effectively handle the given challenge. Additionally, a new knowledge-based operator was created within the context of function evaluation, which focuses on the features of the problem to utilize machine downtime to enhance the quality of the solution. To produce the offspring, a machine-based crossover with order-based precedence preservation was suggested. Additionally, a neighborhood search strategy based on simulated annealing was utilized to enhance the algorithm's capacity for local exploitation, and to broaden its usability. Numerous examples were gathered from the Thailand Tourism Department to demonstrate the effectiveness and efficiency of the proposed approach. The proposed hybrid genetic algorithm's computational results show good performance. We found that the hybrid genetic algorithm can effectively generate a satisfactory tourism service, and its performance is better than that of the genetic algorithm.

**Keywords:** service scheduling; hybrid genetic algorithms; simulated annealing algorithms; tourism services; sustainability tourism

## 1. Introduction

In this era of disruption, people around the world are utilizing information technology to create competitive advantages, especially in the tourism industry. This approach should provide the best solutions and prompt responses that will improve experiences and enhance suitability tourism. Since scheduling is involved in tourism service applications, it is

regarded as a crucial topic in production management and combinatorial optimization. One of the most significant subjects with a wide range of applications in tourism services is scheduling. When assigning resources to a collection of tasks, scheduling must take time, capability, and capacity limits into consideration [1]. To increase profit, the fundamental goal is to increase production efficiency and resource use. Many scheduling issues in the industrial sector are regarded as being extremely complicated, making it challenging to resolve them using precise techniques and standard algorithms. Since 1950, scheduling issues have drawn the attention of several scholars, and significant research efforts have been made in a variety of engineering and scientific domains, including operations research, computer science, industrial engineering, mathematics, and management science.

Scheduling problems are well-known, significant, and difficult issues in the domains of combinatorial optimization and production management. The complexity of scheduling difficulties can be estimated in relation to all potential schedules, and as the size of the problem increases, this complexity increases dramatically.

Scheduling issues are among the hardest problems, and Garey et al. [2] and Ullman [3] demonstrated that this means that polynomial time methods cannot be used to solve them. In cases of scheduling issues have primarily been addressed using traditional approaches and exact methods, such as Lagrangian relaxation, branch and bound, heuristic rules, and shifting bottleneck (e.g., Carlier and Pinson [4], Adams et al. [5], Vancheeswaran and Townsend [6], Brucker et al. [7], and Lageweg et al. [8]). Various techniques have drawn inspiration from nature, biology, and physical processing over the last ten years. These methods have been effectively used to solve a variety of optimization issues, including scheduling issues. Genetic algorithms [9–11], ant colony optimization [12], imperialist competitive algorithms [13,14], tabu search [15], simulated annealing (SA) [16,17], particle swarm optimization [18], and immune systems [19] are a few of these metaheuristic approaches. For further information, see the thorough studies of scheduling problem solutions performed by Jain and Meeran [20] and Ali³ and Bulkan [21].

A potent search method that mimics biological evolution and natural selection is the genetic algorithm (GA). Holland [22] was the first to come up with the idea for GA, and David Goldberg expanded on it. In contrast to other algorithms, this metaheuristic approach is frequently used to locate optimal or nearly optimal solutions to a variety of optimization problems. Davis [23] was the one who first used GA in the scheduling problem, and several GA-based methods for solving scheduling problems have since been introduced. A GA for solving scheduling problems with an encoding technique based on preference rules was proposed by Croce et al. in [11]. A GA with a novel representation technique based on operation completion time was proposed by Yamada and Nakano [24], and its crossover can provide active schedules. For the scheduling problems, Lee et al. [25] presented a GA with an operation-based representation and an order-based crossover that preserves precedence. For scheduling problems, Sun et al. [26] created a modified GA with clonal selection and a life span strategy. Out of 23 key instances, the generated algorithm was able to discover the 21 best solutions. The literature makes it clear that GA is a potent search approach with strong global search capabilities; yet, this metaheuristic algorithm suffers from early convergence and poor local search capabilities.

The major goal of the hybridization strategy is to improve the efficiency and potency of GA by addressing its limitations in terms of local search capability and premature convergence. In their hybrid optimization technique, Wang and Zheng [27] combined local search and GA. The hybrid framework of this method lowers the parameter dependence of both algorithms and reduces the likelihood of GA becoming stuck on local optima. In order to solve the scheduling problems using an adaptive mutation operator and avoid premature convergence, Zhou et al. [28] created a hybrid heuristic GA. In this approach, heuristics were utilized to identify the remaining operations in the constrained solution space, while GA was applied to the machines' first operations.

In order to further boost the quality of the hybrid heuristic GA-derived solutions, a neighborhood search strategy was used. A deadlock-free local search GA with an operation-

based representation and UOX crossover was presented by Ombuki and Ventresca [29], and this was able to produce workable solutions. In this technique, a local search operator was used for local exploitation, while the GA was used to perform global search. They also created a second hybrid genetic algorithm that uses tabu search; according to computational results, the hybrid GA with tabu search performed better than the local search GA. Gonçalves et al. [30] combined GA, a schedule builder, and a local search operator to create a hybrid GA for addressing scheduling problems. In this approach, a priority rule was utilized to produce schedules, and GA was employed to establish priorities. The quality of the solutions was then further enhanced using a local search operator. A hybrid approach using the novel representation technique known as random keys encoding was created by Lin and Yugeng [31]. In this approach, a neighborhood search was added to perform local exploitation and improve the quality of the solution acquired by GA, after the GA was used to obtain an optimal schedule. A hybrid GA was suggested by Zhou et al. [32] to reduce weighted tardiness in the scheduling problem. The first operations of this algorithm were determined using GA, and the remaining operations were assigned using a heuristic. This method was utilized to produce optimal solutions. The hybrid framework outperformed GA and heuristic used alone, according to the results. Agents were employed to implement the parallel GA that Asadzadeh and Zamanifar [33] proposed, as well as to establish the initial populations. A hybrid SA immune system algorithm was presented by Zhang and Wu [17] for reducing the overall weighted tardiness of workshop scheduling. A hybrid micro-GA created by Yusof et al. [34,35] was used in scheduling problems. This method combined an autonomous immigration GA with subpopulations with an asynchronous colony GA, which had colonies with low populations. This algorithm is suitable for use in the complex optimization problem, and can also be applied in tourism service scheduling.

The problem addressed in this research is to find effective algorithms for a tourism recommendation system in Thailand. For the purpose of solving non-preemptive scheduling problems, an efficient hybrid GA is described in this study. An operation-based representation is utilized to more efficiently address the given problem. Additionally, the context of function evaluation involves the invention and use of a new knowledge-based operator. In order to determine the operations of machines during their downtime, this knowledge-based operator mimics the features of the scheduling problem. In order to create offspring and mutants during the reproduction phase of the GA, two different mutation operators and a machine-based, precedence-preserving and order-based crossover are devised. Additionally, SA is used to some extent to increase the population variety in GA, as well as to improve the solution quality of the schedules obtained from GA. After highlighting its key components, the suggested approach is designed to reduce time of schedules. Finally, the computational outcomes of the given issues are given to demonstrate the effectiveness of the suggested algorithm.

To bridge the gap in knowledge regarding the research problem, the aim of this study was to develop a hybrid genetic algorithm by combining a genetic algorithm and a simulated annealing algorithm using a gradient search method to optimize the complex processes used to deal with the tourism service problem, and to compare this to the traditional genetic algorithms used in Chiang Mai smart city case studies in Thailand. The remainder of this paper is structured as follows: The scheduling problems are described in Section 2, while Section 3 contains the suggested hybrid GA. In Section 4, the computational results of exemplar instances are shown, followed by the discussion. Section 5 provides the reasons for choosing the GA and SAA algorithms as effective approaches, then we give our recommendations for tourism systems, ideas, conclusions and proposals for further research.

## 2. Materials and Methods

### 2.1. Tourism Pacagke Tours and Service Scheduling

Package tours, according to Enoch [36], are a useful and more economical way to see tourist locations, but they lack flexibility because of their predetermined itinerary. The

advantages of package vacations frequently exceed their drawbacks for travel agents and customers. Travelers can save money, for instance, when travel agents offer them a discount due to bulk purchasing. Additionally, through the tour operator, the travel firms handle every aspect of travel, transfers, and lodging. By making just one call to confirm their package trip, passengers can save a lot of time compared to having to communicate with several service providers separately. Due to the benefits they provide, package tours are frequently chosen by customers as their means of departure.

Tour operators cater to particular customer groups and provide a variety of excursions, including fully escorted, adventure, special interest, city or regional, and group trips. GPTs are frequently used by citizens of Asian nations and regions, including Taiwan, Japan, South Korea, Hong Kong, and China, as one of their preferred outbound modes of transportation [37]. These trips are now crucial for the growth of individual nations' economies, as well as the global tourist industry [38].

With a wealth of literature on GPT services, previous studies on GPT management have focused on issues with passenger happiness, consumer behavior, and customer service [39]. The practice of combining airline seats and beds in hotels (or other forms of accommodation), in a manner that will make the purchase price acceptable to potential vacationers [39], is termed a package holiday. Seddighi and Theocharous [40] established a framework for analyzing the nature, shape, and character of vacation decision-making, as well as a micro-econometric approach to individual travel. The topic of vertical integration in the European tourism sector was investigated by [41]. Jin et al. [41] looked into how customer decision behavior in similar package tours was affected by the selection framework for upgrading and downgrading. From the viewpoint of the asymmetry in tourism information, Ref. [42] addressed the causes of low tourism quality.

Alao and Batabyal [43] indicated that the characteristics associated with the problem of how to sell package tours to tourists include asymmetric information, uncertain demands, and differentiated customers. Taking these three factors into account, they applied the first contract–theoretic analysis in order to provide the sharing of valuable information between package tour-selling firms and tourists. Aguiar-Quinatna et al. [44] analyzed customers' motivations and perceptions, and proposed strategies to improve the competitiveness of traditional travel agencies in Spain.

Seddighi and Theocharous [45] developed a framework for analyzing the nature, shape, and personality of vacation decision-making, as well as a micro-econometric approach to individual travel. The subject of vertical integration in the European tourism sector was examined by Theuvsen [46]. Jin et al. [47] looked into how client decision-making in similar package trips is impacted by the choice framework for upgrading and downgrading. From the standpoint of information asymmetry in the tourism business, Chen et al. [48] investigated the causes of low tourist quality.

Batabyal and Yoo [49] used probabilistic analysis to determine the long-run loss in demand for guided tours during the low season by modeling the pattern of client arrival as a stochastic process. In their investigation of the UK package tour market in 2007, Davies and Downward examined the dynamics of profitability and market share, as well as the function of pricing and non-pricing decisions, in order to identify the causal mechanisms behind both pricing and competition. In order to examine how Norwegians' income levels affect their demand for inclusive tour charters during a specific time period, Jrgensen and Solvoll [50] created an econometric model. Some tour providers are also focused on offering package holidays or single-destination included tours (Tepelus) [51]. However, to the best of our knowledge, no research has looked at tour selection and bus planning. A number of specific studies have also been conducted on themes relating to package tour management. Portfolio management, facility location selection, supplier selection, and investment portfolio management are examples of issues that are somewhat linked to our research (e.g., Arabani and Farahani [52]).

There is no requirement to take into account whether a preset minimum demand or load is met when assessing a reward or service in relation to these issues. Revenue

is counted once a product has been purchased—for instance, in the product portfolio management problem. When an investment is sold off in the portfolio investment problem, a profit or loss is realized. If a facility can deliver a service to a geographic area within a given distance, the facility location-selection problem's requirement is satisfied. The issue with staff assignments includes any service provided to a customer by an allocated employee. These observations help us understand how the proposed selection differs from other selection-related problems.

The proposed selection problem relates to the minimal number of passengers, whereas other challenges do not. This is the first distinction between the tour portfolio and other portfolio selection problems. The second difference is that while lost sales in most industries are caused by excessive demand, they can also result from insufficient demand in the tourism sector. In other words, new reservations are only turned down once the number of customers who have made reservations has reached capacity.

However, if there are fewer than the required number of consumers for a tour, all reserved clients are refused the service product. This issue is crucial for the tourism sector, because bad proposed selection and bus planning can result in financial losses and a negative business image. There is a research gap in the analysis of proposed selection and bus planning as a result.

Our research offers a mathematical model to look at the particular issues so as to close this gap in the literature. Through the use of a mathematical model, travel companies can profit from quicker and more accurate decision-making by analyzing the costs and benefits via data from the actual world. Travel agencies can utilize the selection method to arrange their package tours and their sightseeing buses, which can provide insights into how to do so. Because they are computationally challenging, the majority of portfolio selection problems that fall under the category of combinatorial optimization problems and traditional mathematical techniques frequently fall short of producing ideal or promising solutions. Due to their high computational efficiency, meta-heuristics-like genetic algorithms have been effectively used to solve real-world combinatorial optimization issues (e.g., Luan et al. [53]; Niknamfar and Niaki [54]). This article also suggests a hybrid heuristic based on using a genetic algorithm to solve the suggested mathematical model in order to accelerate computation.

### 2.2. Tourism Service Scheduling in Smart City

A smart city is a city with physical characteristics, and complex social and digital innovation and information technology. The concept of a smart city refers to a city that is as physically and socially connected as it is digitally [55]. Since the beginning of smart city development, the smart tourism city has emerged as a new concept for utilizing information technology to support the tourists and also the tourism industry in a city [55,56].

In Thailand, Chiang Mai is one of the smart cities that was developed in 2017 to build smart agriculture, smart economy, smart safety, smart health and smart tourism. The policy of smart tourism, which implements ICT infrastructure and technology to serve tourists, aims to support good living and the provision of useful information to people in the area, and also support the tourism industry [57].

In order to improve the competitiveness of the tourism industry, many scholars have addressed the problem of tourist trip design and tour route planning. An attempt to schedule tour guides for the tourism and leisure industry was carried out by Chen and Wang [58]. Their findings demonstrate that the Genetic Algorithm is useful for tourism scheduling service centers. Furthermore, Thumrongvut, Sethanan, Pitakaso, Jamrus, and Golinska-Dawson [59] created a mobile application using a mixed-integer linear programming approach for small-size problems, to help the suppliers of tourism services arrange trip and route planning for visitors, in order to improve the timeliness of visitor services.

### 2.3. Hybrid Genetic Algorithm

There is still much to learn about the extent of the flow shop scheduling challenge established in earlier investigations. Initializing the population will enable us to begin the process in GA. Numerous methods have been employed due to the significance of this process, including opposition-based (Li and Yin, [60]), quasi-oppositional, space transformation searches, and linear regression (Hassanat et al.) [61].

Due to a flow's increased complexity, metaheuristics must be used instead of the traditional heuristic approach (Lee and Loong) [62]. Instead of a memetic or hybrid scheme, a single algorithm dominates the majority of the previous studies (Tosun et al.) [63]. The first metaheuristic used to solve shop flow was simulated annealing (Osman and Potts). GA (Rahman et al.) and the well-known local search optimization technique TS were the next two (Ben Cheikh-Graiet et al.) [64]. Integrating the GA and TS is an alternative technique of solving shop flow, since the hybridization concept provides a more reliable method by combining two or more algorithms. The reason for choosing the GA and SAA algorithms is that they are effective in verifying our recommendations for the tourism system.

Some hybridized algorithms, particularly metaheuristic algorithms, have been used to tackle optimization issues, such as GA, which has already been merged with TS via a scatter search mechanism (Glover et al., 1995). There is the general algorithm (GA) that has been hybridized with the insertion of the search cut and repair algorithm (GA-ISCR) (Tseng and Lin, 2010), the co-evolutionary quantum genetic algorithm (CQGA) (Deng et al. [65]), and the general algorithm combined with simulated annealing (HGSA) (Wei et al.) [66].

In addition to GA, a number of different metaheuristics have been proposed to decrease the life span of shop flows, including those by Ding et al. (2015), who developed a local search approach using variable neighborhood search, called the tabu-mechanism with improved iterated greedy algorithm (TMIIG); there is also the iterated greedy with jumping probability (IG-JP), developed by Tasgetiren et al. [67], and Davendra and Bialic-Davendra (2013) developed DSOMA. Additionally, the aquila optimizer, which is based on how an aquila behaves when hunting its prey, has been applied in general optimization problems in recent years (Abualigah et al.) [68]. CFD analysis and the optimization of the design of a centrifugal pump use an effective, artificially intelligent algorithm that optimizes the compliant gripper mechanism's design by employing an effective bialgorithm comprising fuzzy logic and ANFIS, which is an efficient hybrid approach to the finite element method. The artificial neural network-based multi-objective genetic algorithm has also been used for the computational optimization of a linear compliant mechanism of a nanoindentation tester.

Wang et al. (2022) [69] proposed a cutting-edge strategy to raise the efficiency, power, and head of centrifugal pumps. First, an ideal numerical model that took pump efficiency and head into consideration was explored in order to set restrictions. Before conducting testing, the pump was built, and an artificial intelligence algorithmic technique was applied to the pump. By choosing the parameters for the centrifugal pump's casing section area, the impeller interference, the volute tongue length, and the volute tongue angle were investigated in a number of models. On the optimization indices, the weights of the safety and displacement factors were estimated. The weights matrix for the ideal process ranged from less than 38% to more than 62%. This method ensures that the optimization approach is strong and multi-objective. The outcomes demonstrate an improvement in centrifugal pump performance.

### 2.4. Simulated Annealing Algorithm

The original single-objective simulated annealing algorithm was slightly modified by Serafini (1994) by changing the method of calculating the probability of accepting new candidate solutions to narrow the search. This was one of the first works to use a simulated annealing algorithm to solve multi-objective problems. In addition to the well-known traveling salesman issue, a smaller problem with two objectives was also solved using the

approach. The technique performs well for minor issues, but with large problems, only limited parts of the Pareto front can be reached, as claimed by Czyzak and Jaszkiewicz [70].

In order to find a good approximation of the entire Pareto front in somewhat large multi-objective knapsack problems with two, three, and four objectives, they invented Pareto-simulated annealing (PSA). In an effort to ensure the dispersion of the nondominated solutions discovered throughout the procedure, PSA employs objective weights that are changed after each iteration. In order to solve the two-objective knapsack problem, Ulungu et al. [71] proposed the UMOSA (Ulungu Multi-Objective Simulated Annealing) approach and investigated the impact of various neighborhood architectures (for candidate solution generation) and aggregation functions. All of these older MOSA techniques combine the goals using a variety of aggregation functions and objective weights, and the aggregated values are then minimized, much like in the case of a single simulated annealing process. Although it is necessary to create objective weights in advance, doing so can be difficult because, as Das and Dennis [72] found when using a number of instances, inadequate weight values render some areas of the Pareto front unreachable.

Following these initial attempts, several academics began utilizing the dominance idea to direct the algorithm. SMOSA (Suppapitnarm multi-objective simulated annealing), a technique developed by Suppapitnarm et al. in 2000, uses nondominated solutions that are archived in a solution list. The nondominated solutions discovered during the procedure had to be saved in an archive because SMOSA only produces one solution in each iteration (as opposed to MOEAs algorithms, which deal with populations of solutions). It employs a cutting-edge "return to base" tactic. In order to repeat the search and attempt to reach a new area of the front, the return to base approach occasionally chooses a random solution from the archive. Sankararao and Yoo [73] claim that even with this revised formulation, few nondominated solutions are found, and the convergence velocity is slow due to the search method employed. The acceptance determination process for this algorithm is primarily based on direct comparisons between the candidate and existing solution, which means that several excellent solutions may be overlooked while searching.

In his 2003 study, Suman [33] compared the performance of the UMOSA, SMOSA, and PSA algorithms, as well as two new ones: the PDMOSA (Pareto dominance multi-objective simulated annealing), which uses the Pareto dominance concept to direct the search, and the WMOSA (weighted multi-objective simulated annealing), which uses a weight vector to handle the constraints that are not verified. Certain comparisons aim to demonstrate how effective these algorithms are at addressing multi-objective optimization problems.

All of the approaches reached nondominated solutions, according to the comparisons, although the PDMOSA results are subpar compared to those of the other methods, and in some cases, PSA performs best. To assure a large collection of nondominated solutions, the authors contend that all five techniques need little processing effort and should be applied simultaneously. This means that if each of these methods is utilized independently, a larger number of front-end solutions cannot be guaranteed.

Suman and Kumar [74] reviewed SA-based algorithms and suggested new lines of inquiry to enhance MOSA algorithm performance. Suman et al. [75] proposed the orthogonal simulated annealing (OSA) technique in order to avoid utilizing conventional approaches that could inadvertently choose a new candidate solution at random, rendering the algorithm ineffective in exploring the solution space in complicated optimization problems. To increase the algorithm's convergence, they suggested employing methods to first search the space of potential candidate solutions in the vicinity of the current solution, and then pick the best candidate solution. However, when compared to other algorithms, OSA's performance was problem-dependent; while there are some situations in which OSA succeeds, there are other instances in which it fails. It is essential to develop an algorithm that performs well in a wide range of circumstances.

Bandyopadhyay et al. [76], who pioneered the archive multi-objective simulated annealing (AMOSA) approach, made a significant advancement in this subject. To calculate trade-offs between solutions, this algorithm makes use of an archive that holds the nondom-

inated solutions discovered during the optimization process. Additionally, they devised a process that uses the notion of the quantity of domination and the domination status amongst solutions to direct the algorithm search.

AMOSA's results were superior to those provided by NSGA-II (Deb et al.) [77] and PAES (Knowles and Corne) [78], in that a greater number of nondominated solutions were obtained, according to the results of various tests. The archive taken into account by AMOSA has two separate size restrictions, with an upper limit on the number of nondominated solutions that can be stored. The number of answers is then reduced to a smaller limit using a clustering strategy, which involves controlling and separating a huge amount of data into groups or subsets with comparable characteristics. This lowers the algorithm's computing requirements.

No constraints should be taken into account, however, to arrive at a more varied collection of nondominated solutions. Furthermore, as Sankararao and Yoo [79] did not seek a process to obtain a uniform Pareto front, it was discovered that the large number of solutions obtained by AMOSA were concentrated in specific regions of the front. Sankararao and Yoo [80] developed the multi-objective simulated annealing technique (rMOSA). This is a process to select the "most uncrowded solution" in the archive to generate a new candidate solution and obtain a well-distributed number of solutions in the algorithm's final results. This process seeks to select a random solution from the archive to generate a new candidate solution, with the aim of increasing the convergence speed of the algorithm.

The optimization process begins with the simultaneous employment of these search techniques, and they are applied periodically during the iterations. On the basis of comparisons with the existing solution, rMOSA employs a straightforward process for accepting candidate solutions. This can be ineffective, since it is important to consider the nondominated solutions that have already been found when accepting new candidate solutions (Cao and Tang) [81]. Additionally, the result derived when utilizing both processes concurrently from the start ignores the algorithm's evolution.

In an effort to improve the intensification and diversity aspects of the UMOSA algorithm, Rincón-Garca et al. [82] created a variation of the algorithm. They proposed various aggregation strategies for meeting the aims. The fundamental drawback of this form of weight-based algorithm is still, however, the establishment of weights allocated to targets in advance.

To address issues in air traffic control systems, Mateos and Jiménez-Martn [83] suggested a better AMOSA. To calculate the likelihood of adopting inferior solutions in comparison to the original AMOSA scheme, a new "element" was proposed. By only accepting solutions that increase the variety of the front, this element is utilized to speed up the algorithm's convergence rate. The authors showed that the algorithm performs well, but they contend that in order to demonstrate its superiority, it should be compared to other algorithms that have been applied to the same problem. It is important to carefully consider throwing out some answers that would increase the convergence velocity.

Sengupta and Saha [84] introduced a novel unconstrained many-objective optimization technique that was evaluated on academic optimization issues—the reference point-based simulated annealing (RSA) method. In contrast to the earlier "point-to-point" MOSA algorithms, the authors suggested an "archive-to-archive" procedure, which greatly improves the outcomes. The authors demonstrated that RSA performs effectively in many types of problems, based on comparisons with the outcomes of other methods.

Similar to the AMOSA algorithm, RSA uses clustering to shrink the archive, and as previously mentioned, this should be carefully considered to maintain the variety in the final set of answers. The AMOSA method is also used by Saadatpour et al. [85] to tackle a multi objective waste load allocation problem in rivers. In this instance, the archive is designed to hold just a small number of nondominated solutions. As a result, the finished front's quality may suffer.

Our work introduces a novel MOSA-GR technique, which was motivated by the encouraging outcomes from a wide range of applications of MOSA algorithms. The new

method was developed by using the finest ideas from the MOSA algorithms already mentioned above, and avoiding their shortcomings, notably with regard to the convergence and spreading of solutions along the front. The procedure for creating new candidate solutions and the reannealing of the algorithm are the two key areas where we suggest new enhancements to a previous version that Marques et al. [86] already provided. By utilizing a special tool that excels in a wide range of situations, this novel method seeks to identify a nearly optimal and sizable collection of nondominated, well-dispersed solutions.

Furthermore, there are two studies related to our research that apply an optimization technique in other areas of industry; for example, Zhang et al. (2018) [87] evaluated trade-offs between the main factors in green manufacturing (energy, noise and cost) through cutting parameter optimization, which introduces the new optimization technique to the manufacturing industry. Moreover, Zhang et al. (2020) [88] studied a multi-objective cellular genetic algorithm for the energy-oriented balancing and sequencing problem in a mixed-model assembly line in the energy industry. Finally, many algorithms have been designed to solve multi-objective optimization issues in the literature. However, the creation of new algorithms continues to be crucial. New algorithms are required to solve these problems in an acceptable amount of time, because the complexity of the problems is continuously increasing.

## 3. Methodology

### 3.1. Simulated Annealing Algorithm

Problem Definition

The JSSP can be expressed as follows: JOB SHOP SCHEDULING PROBLEM. Each of the "$n$" jobs consists of a number of procedures that must be carried out on "m" different machines. A single machine is employed for a set amount of time for each operation. Each operation can only be processed by one machine at a time, and processing must continue uninterrupted until completion once processing on a given machine begins. A specific job's processes must be completed in a specific sequence. The fundamental goal of the challenge is to plan the activities or operations of the tasks on the machines so that the overall time to complete all of the jobs, or the makespan (Cmax), is as short as possible. The phrase "makespan" describes the total amount of time needed to finish all operations for all tasks on all machines. It is a measurement of the amount of time elapsed between the beginning of the first operation and its conclusion. Sometimes, there may be several solutions with the shortest makespan, but the objective is to identify any one of them; finding all feasible optimum solutions is not always necessary.

The scheduling problem is a broad kind of the following classical scheduling problems:

given tourists $n$, $N = \{j1, j2, \ldots, jn\}$;
given tourists consists of $s$ operations $S = \{Oj1, Oj2, \ldots, Oj,s\}$.

Each tourist gives rise to tasks that need to be completed on services in a specific technological order: $S = \{Oj1, Oj2, \ldots, Oj,s\}$. The notation indicates the third operation of a project that must be completed on one of the services and has a known processing time. In this setting, each computer can only handle one operation at a time, and a single tourist's operation cannot be handled by two services concurrently. Operation $Oj,s$ must be processed on one of the designated machines until it is finished, without being interrupted. Additionally, a job can only visit one machine once; subsequent visits to the same machine are not permitted. Additionally, it is believed that the machines are always available and that operation travel times are minimal. In contrast to flow shop scheduling, scheduling problems assign each job a certain planned route. The order of all activities on all machines is planned to reduce Cmax, or the maximum time for a project to be completed (max $C1$, $C2, \ldots, Cj$).

A scheduling problem uses a mathematical model with the goal of reducing the makespan. In this model, B is taken to be a large integer, where $tj,s$ represents the start of

operation $O_{j,s}$, $Sm_{i,l}$ represents the start of machine I in the priority of $l$, and $h_{i,j,s}$ is given a value of 1 if operation $O_{j,s}$ is carried out on machine I, and a value of 0 if otherwise:

$$\min Z = \max \{C1, C2, \ldots, Cj\}\ j = 1, 2, \ldots, n \tag{1}$$

$$S.t.\ t_{j,s} + P_{j,s} \le t_{j,s} + 1\ j = 1, 2, \ldots, n;\ s = 1, 2, \ldots, s_j - 1 \tag{2}$$

$$Sm_{i,l} + P_{j,s}X_{i,j,s,l} \le Sm_{i,l} + 1\ i = 1, 2, \ldots, m;\ j = 1, 2, \ldots, n;\ l = 1, 2, \ldots, l_i - 1; \\ s = 1, 2, \ldots, s_j \tag{3}$$

$$Sm_{i,l} \le t_{j,s} + (1 - X_{i,j,s,l})\ B\ i = 1, 2, \ldots, m;\ j = 1, 2, \ldots, n;\ l = 1, 2, \ldots, l_i; \\ s = 1, 2, \ldots, s_j \tag{4}$$

$$X_{i,j,s,l} \le h_{i,j,s}\ i = 1, 2, \ldots, m;\ j = 1, 2, \ldots, n;\ l = 1, 2, \ldots, l_i;\ s = 1, 2, \ldots, s_j \tag{5}$$

$$\sum_j \sum_s X_{i,j,s,l} = 1\ i = 1, 2, \ldots, m;\ l = 1, 2, \ldots, l_i \tag{6}$$

$$\sum_i \sum_l X_{i,j,s,l} = 1\ j = 1, 2, \ldots, n;\ s = 1, 2, \ldots, s_j \tag{7}$$

$$t_{j,s} \ge 0\ j = 1, 2, \ldots, n;\ s = 1, 2, \ldots, s_j \tag{8}$$

$$X_{i,j,s,l} \in \{0, 1\}\ i = 1, 2, \ldots, m;\ j = 1, 2, \ldots, n;\ l = 1, 2, \ldots, l_i;\ s = 1, 2, \ldots, s_j. \tag{9}$$

Constraint (2) in this model deals with the sequences of operations that must occur in a particular order. The third constraint forbids machine overlap and mandates that no machine process more than one operation at once. Constraint (4) forbids operations from running simultaneously; therefore, an operation is assigned to a specific idle machine under the condition that its preceding operation is completed. Additionally, a machine is selected in a constraint for each of the operations (5). Operations are assigned to the machines and are performed in the order specified by constraint (6). The number of operations that can be carried out on a single machine is constrained by constraint (7), based on the priority of the machine.

*3.2. The Proposed Hybrid Genetic Algorithm-Simulated Annealing Algorithm*

Even though the proposed hybrid genetic algorithm–simulated annealing algorithm is a traditional algorithm, it still works effectively for our recommendation system. The hybrid algorithm that is being suggested combines the GA and SA algorithms. The benefits of both algorithms are combined in the suggested framework to get the best answers to the scheduling problems. A flowchart for the suggested hybrid genetic algorithm (HGA) is shown in Figure 1. The random initialization of the population is where it all begins. A new knowledge-based operator is used in this stage to enhance the individual's solution quality, as the formed population is assessed using the fitness function. This knowledge-based operator also integrates with the function assessment stage and operates during machine downtime.

A selection operator is used during the reproduction phase to choose the mating pool's parents, followed by a crossover to create the progeny, when the operator is used. To create the mutants, a mutation operator is also applied to a group of people chosen at random. The evaluation of the produced offspring and mutants is followed by consideration of the termination circumstances, which directs the algorithm in the appropriate direction. Section 3.7 details the termination requirements.

However, the simulated annealing algorithm will converge to a fixed point in simulated annealing; the acceptance probability for a new state in step $k$ is traditionally defined as

$$(accept\ new) = \begin{cases} exp\left(-\frac{\Delta}{T_k}\right) & , if\ \Delta \ge 0 \\ 1 & , x < 0 \end{cases}$$

where $\Delta = f(new) - f(old)$ is the change in objective function $f$, which is to be minimized, and $T_k$ is a strictly decreasing positive sequence with $\lim_{k \to \infty} T_k = 0$.

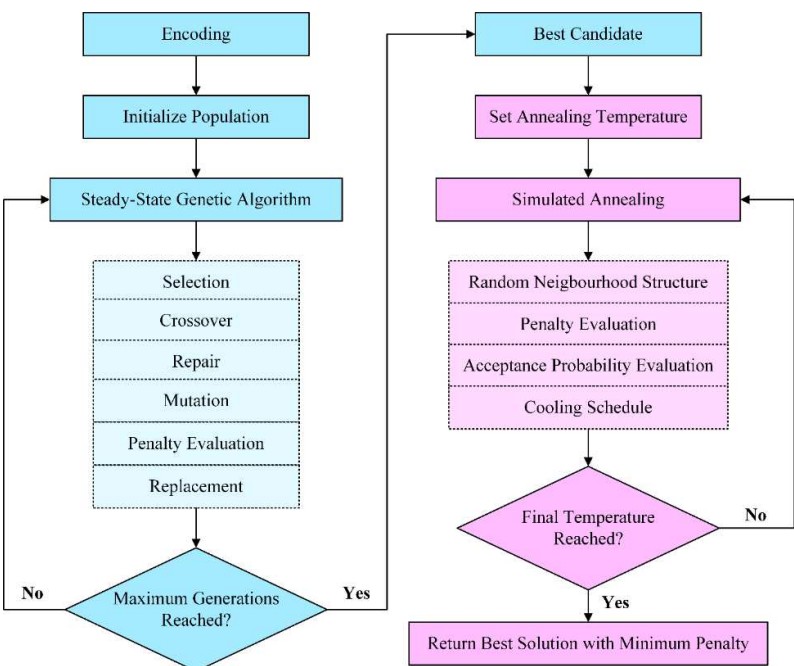

**Figure 1.** The proposed hybrid genetic algorithm–simulated annealing algorithm.

When the algorithm runs through the termination condition, which is when GA's generations come to an end, SA will begin with the top GA performer. In SA, a neighborhood search structure is used with three separate operators as the proposal mechanism: swapping, insertion, and reversion. Additionally, as each SA cooling criterion is met, a pool of people with the beta percent of accepted solutions is maintained.

In Section 3.6, the SA algorithm is covered in detail. Three conditions are set in this phase of the algorithm to either stop it or proceed with fresh parameters. The parameters of SA are reset and fresh GA parameters are applied if the most recent condition, or condition 4, which is the termination of SA's outer loop, is reached. Additionally, from the migration pool of SA, zeta percent of unique persons are relocated to GA.

*3.3. Encoding and Decoding*

Finding the right encoding and decoding techniques to describe the problem is the first and most crucial stage in any algorithm. The permutation of operations for various occupations is represented in this paper using an operation-based representation [34,37]. If the technology limitations are met, the timetable could be built using this representational approach. According to this method, a chromosome is made up of nm genes, each of which encodes the order of operations to be carried out on the machines.

Each operation is indicated by a positive integer number ranging from 1 to $n$ (see in Figure 2). Each of the integer values has an equal number of occurrences and operations. In other words, the kth instance of an integer value in a chromosome corresponds to the kth technological procedure of the task. Each tourist on this chromosome consists of three processes, and as a result, every job appears three times along the length of the chromosome. For instance, this chromosome's sixth and ninth genes, respectively, reflect the second and third operations of jobs one and two. Each chromosome also has other data that are related to it, such as the machine number, processing time, start time, and finish time.

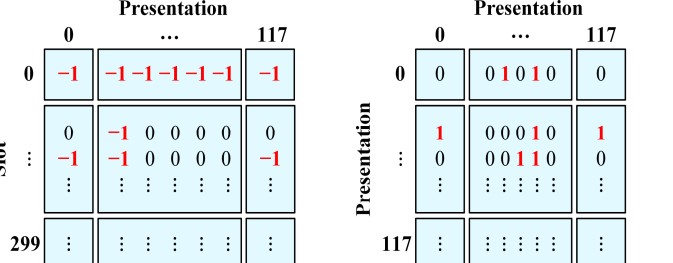

**Figure 2.** From the left, presentation-by-presentation matrix, supervisor-by-preference matrix, and slot-by-presentation matrix.

*3.4. The Knowledge-Based Operator and the Fitness Function*

The likelihood that a solution can be passed on to the following generation in an optimization problem is often determined by the fitness function. In other words, by using this operator, the solution quality is assessed, and the better-fitting chromosomes will have a higher chance of survival; nevertheless, the less well-fitting chromosomes must be eliminated from the population. There are numerous possible performance evaluators for defining the fitness function in scheduling problems. In this study, the fitness function used to assess each chromosome is either makespan or Cmax. A new knowledge-based operator is created in this algorithm and in the context of the fitness function based on the characteristics of the challenge. Based on the machine idle times present in job shop settings, this operator was created. Additionally, in order to shorten the algorithm's computation time and cover all the chromosomes that need to be evaluated, this knowledge-based operator is used during the function evaluation phase. The next stages are used to develop this operator more effectively.

The first step is to locate each machine's idle positions. The idle start time, idle finish time, and idle time must then be determined for each of these inactive points.

Step 2. Taking into account the length of idle time as well as the processing time of the candidate operation, a candidate operation is selected from the right side of the machine sequence list based on the position of the idle point on the machine sequence list.

Step 3: If the amount of idle time is greater than or equal to the candidate operation's processing time, it will be conditionally accepted. If not, it will be turned down.

Step 4. If the candidate operation is not accepted for transfer, the operator returns to step 2 and selects the next operation.

Step 5: To decide whether to accept or reject the shift for the conditionally accepted candidate operation, all processing limitations must be taken into account. For instance, the candidate operation's previous operation must be finished.

Step 6: The candidate operation will be moved to its new location if all of the constraints are met. If not, the candidate operation will remain in its current place.

Step 7. Steps 2 through 6 should be repeated for each machine until the very last action on the machine sequence list.

*3.5. Encoding Selection Operator*

An effective selection method can improve GA's performance by enabling it to find the best solutions more quickly. The most popular operator, the Roulette Wheel selection technique, is used in this study to choose the parents [34]. Additionally, the elitism strategy is used in this selection process to keep the fittest chromosomes for the following generation and stop the problem from getting worse over time.

We determined the likelihood of each chromosome in Roulette Wheel selection using Boltzmann Probability $P(x) = \exp(f(x))/f(x)$. $P(x)$ in this equation stands for the likelihood of each chromosome, indicating selection pressure, while $f(x)$ stands for each individual's fitness, and $f(x)$ is the fitness of the generation's lowest performer. It should be noted that in order to make the selection pressure independent of the problem scale, $f(x)$ was

added to the original Boltzmann Probability equation. Additionally, each chosen person's normalized probability is stated as P(x)norm = P(x)/max.pop x = 1 P(x).

### 3.6. Crossover

In GA, crossover is the most crucial operator compared to other operators, and is actually the algorithm's foundation. By merging the data from the first and second parents, the crossover operator creates offspring with traits from both parents that may be better or worse than their parents. Additionally, the major objective of this operator is to create viable offspring using parental knowledge that is better.

In this study, a realistic offspring generation method called a machine-based precedence-preserving order-based crossover (POX) is proposed [25]. The POX operator is implemented using the specific steps listed below. Figure 3 show the process of chromosome which in the red frame show the presentation of process by the following step.

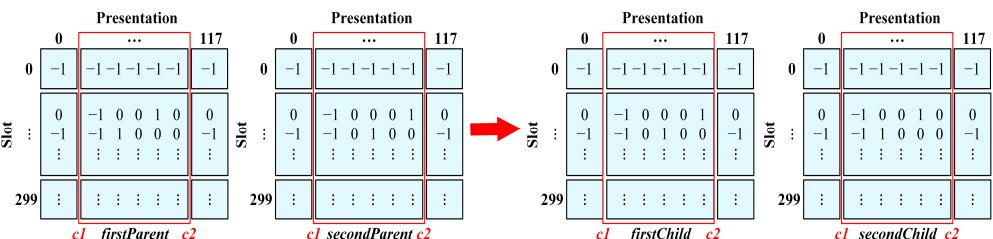

**Figure 3.** Process of a chromosome.

Step 1: Using Roulette Wheel selection, two people are first chosen to serve as parents (P1, P2).

Step 2: Next, two groups of tourists are chosen and given the names sj1 and sj2. One of the tourists is chosen at random from the remaining n jobs, and one is chosen from the bottleneck machines.

Step 3 copies the components of the first tourists (sj1) from the first parent (P1) to the precise locations; in other words, the elements of the second tourists (sj2) are replicated exactly from the second parent (P2) to the alleles in the second child (O2), and the same is true for the second tourists (sj2).

Step 4: In the first tourists (sj1) and the second tourists (sj2), all of the alleles are deleted in the second parent (P2) and, in the case of the second tourists (sj2), all of the alleles are deleted in the first parent (P1).

Step 5: From the furthest left to the furthest right, the second and first children (O2, O1) receive the remaining alleles from the first and second parents (P1, P2), respectively.

### 3.7. Mutation

Mutation is the second method of examining the solution space during the reproduction phase. The mutation operator can make the algorithm faster at finding better solutions and prevent it from getting stuck in the local optima. Additionally, it might alter the chromosome to broaden the diversity of the population.

Two different kinds of mutation operators, namely, swapping and insertion, are used in this approach. The insertion operator could perform a thorough search in addition to increasing the population's variety through mutation. It should be emphasized that one of these mutation operators, which are listed below, should be chosen at random in order to produce an offspring. Figure 4 show the process of chromosome mutation which the red frame shows the result of the process of the following operator.

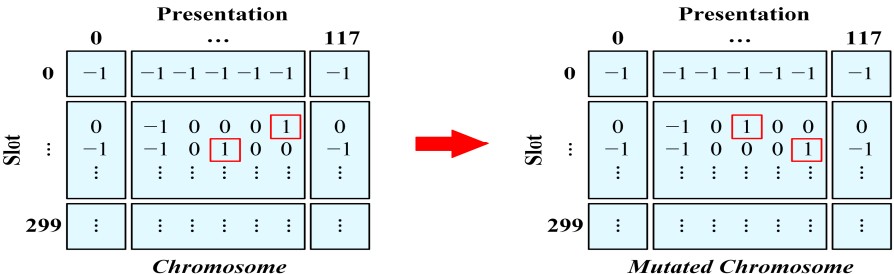

**Figure 4.** Process of chromosome mutation.

(1) Swapping operator: Before using the swapping operator, two random values (R = 2, 5, for example) are created that represent the locations of two alleles on the chromosome. Except for the randomly chosen alleles that must be switched or exchanged in the offspring, all of the parental information is then copied to the exact places in the child. Consider the parent who was chosen at random from the population.

(2) Insertion operator: To use the insertion operator, two random values are first created to represent the locations of two alleles on the chromosome. The offspring's genetic makeup is then identically duplicated from the parent, with the exception of the randomly chosen alleles. The other randomly chosen allele with a greater job-value is positioned to the left of the smaller randomly chosen allele.

Figure 5 show the simulated annealing process to evaluate the accuracy of this algorithm via convergence, the plot of cost vs. iterations makes it simple to see how the SAA is convergent. Other elements with a similar viewpoint include round-off errors and local extrema. In Figure 6, the cost function is the minimum, beginning at generation 1125.

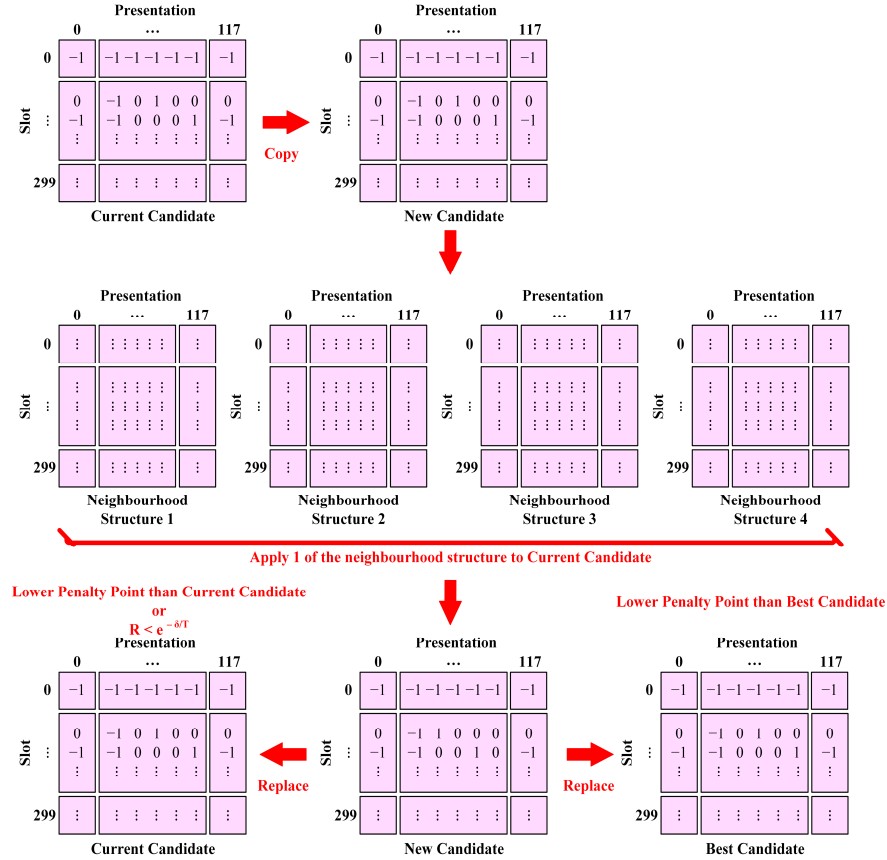

**Figure 5.** Simulated annealing process.

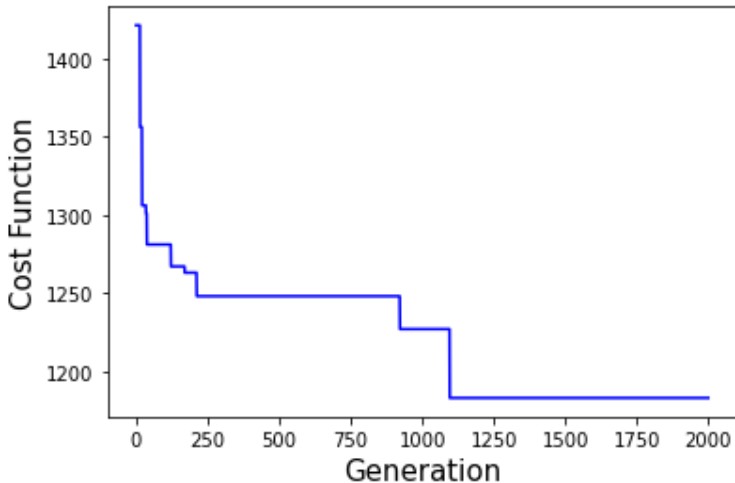

**Figure 6.** Cost function.

## 4. Results

### *4.1. Simulated Annealing*

The SA technique, which is categorized as a stochastic local search approach, was motivated by the physical annealing process of solid materials [35]. In SA, the search operator is periodically allowed to take bad turns, which allows the algorithm to divert from local solutions and move toward global ones. In SA, this trait could be acquired by probabilistically opting for the less desirable options.

In the suggested HGA, SA begins with the top GA solution. Then, using the current solution (S), a proposal mechanism made up of the three operators (swapping, insertion, and reversal) is used to create a new neighborhood solution (S). The freshly created solution (S) is subjected to the new knowledge-based operator, after which it is assessed using the objective function. The newly evaluated neighborhood solution will be accepted (f(S) f(S)) if it is on par with or superior to the current solution. If not, the algorithm will carry on searching for a solution (S or S) determined by a probabilistic acceptance function. Acceptance of a solution is also determined by the value of the individual's goal function and the algorithm's current temperature (T) (e [(f(S)f(S))/T]). An acceptable answer is maintained among a fresh group of people in each inner loop of SA. The starting value of the temperature should be changed in accordance with the annealing schedule as the algorithm's inner loops are ended. Additionally, in each outer loop of SA, the remaining individuals from the new pool of individuals are discarded, while a beta percent of the unique individuals are maintained. Additionally, 25% of unique individuals are retained once the outer loop is closed for migration-related purposes, while the remaining remainders are deleted. It should be emphasized that a low migration rate is required in order to somewhat increase the variety of the GA population. Three operators (swapping, insertion, and reversion) are utilized in SA's proposal method, and one of them is arbitrarily chosen and used to produce the neighborhood solution.

### *4.2. Ending Conditions*

Four distinct conditions are offered in the proposed HGA in order to fully or partially terminate the algorithm. The achievement of the best-known solution serves as the first termination condition, and the maximum number of generations in the HGA main loop serves as the third termination condition. The entire algorithm will be terminated at once if it meets either the first or third criteria. The maximum number of generations in GA and the maximum number of outer loops in SA are the second and fourth partial conditions, respectively.

### 4.3. Computational Experiments and Discussion

The following table provides a comparison between the planned HGA and the traditional GA. The same parameters are used to implement GA and HGA in a Python 3 environment. The fitness and execution time associated with the quantity of convergence iterations serves as the comparison criterion. For the HGA, all parameters are set for the best result after a specific number of iterations (in Table 1).

**Table 1.** Parameters.

| Parameters | Values |
| --- | --- |
| Total number of iterations | 2000 |
| Number of individuals per generation | 50 |
| Chromosomes' size | 8 |
| Mutation probability | 0.2 |
| Mutant gene per individual | 1 |
| Crossover probability | 1 |
| Number of individuals chosen during the selection | 1 |
| Number of iterations for the LS | 6 |
| Number of neighbors per individual | 4 |

The optimization results are compiled in the table. One thousand repetitions later, the ideal value had been reached. By providing a solid resolution, this also demonstrates that all genetic processes function effectively in general. In actuality, the program consistently moves closer to the ideal outcome. HGA, on the other hand, offers a reduction in the number of iterations required to converge from 15 s to 11 s. As a result, HGA can perform better than regular GA on an optimization problem, which demonstrates the hybridization mechanism's ability to improve convergence (in Table 2).

**Table 2.** Genetic algorithm and hybrid genetic algorithm performance.

| | Convergence (Iterations/Time CPU) |
| --- | --- |
| Genetic Algorithm | 15/60 s |
| Hybrid Genetic Algorithm | 11/46 s |

The proposed HGA was implemented using Python, and the technique was tested on a machine running Windows 10, an Intel Core i5 running at 2.5 GHz, with 8 GB of RAM. The results of the computational experiments were derived using the 10 well-researched tourism services with the aforementioned tweaked parameters and 10 replications for each benchmarked instance (Figure 7). The dimension of the problem is tourism services, the best-known solution, and the outcomes of our proposed HGA, which are presented in columns and include the best replication-level solution, relative deviation (RD), average solution, and worst algorithmic solution over ten runs. These are the most crucial indexes to compare in order to evaluate the algorithm's efficiency and consistency.

The notion of the "smart city" has improved the effectiveness and procedure of city management while integrating connection in many different domains. It requires strategic and innovative thinking to implement the smart city program, because it is not simple to do so. The first steps in this strategic program include problem identification, problem grouping, an abstraction process, solution selection, selection of efficient techniques, and implementation planning. The community and organizations can benefit from the innovative and valuable works produced by creativity. It is possible to research a variety of organizational traits, including human resource management and leadership planning. One tactic for raising public service companies' effectiveness is innovation. The limitations in the implementation of smart cities have not stopped efforts to advance numerous applications and other technologies. Chiang Mai has a plan to become a smart city, aiming to improve services to the community by utilizing local knowledge and maximizing the application

of technology. One of the applications is called the Chiang Mai Smart City For Smart Mobility, which stores all the route data in Chiang Mai such that our proposed method can be applied effectively. A smart city is a form of the application of digital technology or information and communication to enhance the efficiency and quality of travel services, and help reduce costs and population consumption. Increasing the capacity of people to achieve a better quality of life, the smart city is a project that many cities around the world have sought to develop, in order to keep up with the new era by using technology. Technology is integrated with the life of the people, whether in terms of transportation power consumption or infrastructure, to make the city more comfortable. Chiang Mai is ready to drive smart city developments via a strategy to improve performance in a systematic way, and thus facilitate the development of a pattern that will drive Chiang Mai to truly become a city of life and wealth.

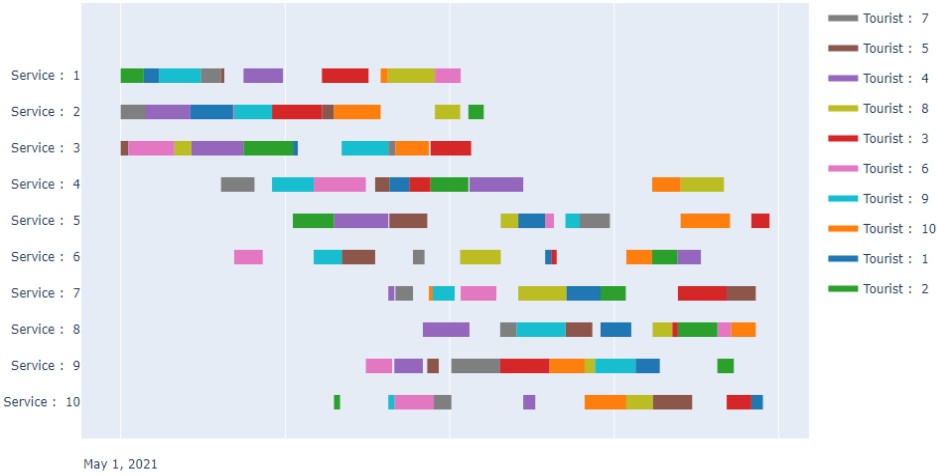

**Figure 7.** Tourism service scheduling.

## 5. Conclusions and Discussion

The algorithm's result was encoded in an operation-based form. The problem features were used to develop a new knowledge-based operator, which was able to improve the schedule solutions' quality. In order to boost population variety and sharpen the search, a machine-based precedence-preserving order-based crossover and two types of mutation operators, namely, swapping and insertion, were used to produce the offspring and mutants. Additionally, the SA method was used to further enhance the GA solution's quality due to its neighborhood search functionality.

The research library's well-studied benchmarked cases were used to test the proposed HGA, and the results were compared to those of other methods.

The usefulness and efficiency of the suggested technique are demonstrated by the computational results, which generally reveal that the proposed algorithm has a lower average relative deviation than the compared algorithms.

This paper adopts a hybrid genetic algorithm–simulated annealing algorithm to create schedules for tourism services, which is meets the interests of active tourists by collecting data from Thailand's tourism industry on the preferences of tourists. Our proposed algorithm is able to create schedules for tourism services and develop recommendations. The system employs minimum costs to create schedules for tourism services by the using genetic algorithm approach. Our proposed hybrid recommendation algorithm focuses on the tourists in Thailand. In future research, this can be extended to compare with other algorithms and improve the recommendation system. The result of this study is a recommended system, related to the study of Duan et al. (2020) [89], for the optimal scheduling and management of a smart city within a safe framework. Moreover, the results show that the hybrid genetic algorithm can effectively generate a satisfaction tourism

service that saves energy which related the study of Alsokhiry et al. (2022) [90]—and thus develops a framework using pricing-based energy management.

A limitation of this study is that the method is time-consuming, which can cause the amount of services to vary if this method is applied in a large area, and will result in a very long time for the result to be output. Moreover, the accuracy of this work will be developed in future research.

We advise future researchers to consider the greenness issue in scheduling, because it is a new field that is expanding in manufacturing sectors. Additionally, recently created algorithms, such as the imperialist competitive algorithm, can be tested using the proposed framework and knowledge-based operator.

Additionally, one might think about creating new operators to further broaden the algorithm's population diversity, and even creating an operator to quantify population diversity.

**Author Contributions:** Conceptualization, P.S. and P.J.; methodology, P.S. and P.J; software, P.S. and P.J.; validation, P.S. and P.J.; formal analysis P.S. and P.J.; investigation, P.S. and P.J, resources, P.S. and P.J.; data curation, P.S. and P.J.; writing—original draft preparation, P.S. and P.J.; writing—review and editing, P.S., P.J., and P.K.; visualization, P.S. and P.J.; supervision, K.J.; project administration, P.S.; funding acquisition, P.S. All authors have read and agreed to the published version of the manuscript.

**Funding:** This research was funded by Suan Dusit University under the Ministry of Higher Education, Science, Research and Innovation, Thailand, grant number 65-FF-003 "Innovation of Smart Tourism to Promote Tourism in Suphan Buri Province".

**Institutional Review Board Statement:** The study was conducted in accordance with the ethical and approved by the Ethics Committee of Suan Dusit University (SDU-RDI-SHS 2022-030, 1 June 2022) for studies involving humans.

**Informed Consent Statement:** Not applicable.

**Data Availability Statement:** Not applicable.

**Acknowledgments:** The research team would like to thank Suan Dusit University for the support with funding. Also, we would like to thank AVIC Research Center, Chulalongkorn University for consulting this paper. Finally, we would like to thank the Chiang Mai Municipality for all their cooperation and for providing the necessary information for the research.

**Conflicts of Interest:** The authors declare no conflict of interest.

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
