# Peer review of "Tourism Service Scheduling in Smart City Based on Hybrid Genetic Algorithm Simulated Annealing Algorithm"

_sustainability, doi:10.3390/su142316293_

Round 1

Reviewer 1 Report

This study provides the calculus for a new approach to scheduling and time planning. In today's smart cities, it can show the cost loss to the tour provider and provide tourists with a more immediate itinerary to reduce the loss of time. The collation of the research literature and the calculation method is appropriate, and the overall calculation results are properly explained. The results of the application are also clearly stated in the use of the Thailand data. However, the sentences need to be restructured, and punctuation should be paid attention to.

Author Response

This study provides the calculus for a new approach to scheduling and time planning. In today's smart cities, it can show the cost loss to the tour provider and provide tourists with a more immediate itinerary to reduce the loss of time. The collation of the research literature and the calculation method is appropriate, and the overall calculation results are properly explained. The results of the application are also clearly stated in the use of the Thailand data. However, the sentences need to be restructured, and punctuation should be paid attention to.

Answer:   Thank you so much for the wonderful reviews and give recommendation for improving to be a better quality paper.

  1. We rewrite some sentences and punctuation to be more effective for communication (Line 20-25; 122-147; 276-278; 292-308; 443-475; 487-488; 503-509; 644-647; 774-783; 938-929; 987-983).

  1. We add the literature that related to this paper as following:

  1.  

[68] Wang C-N, Yang F-C, Nguyen VTT, Vo NTM. CFD Analysis and Optimum Design for a Centrifugal Pump Using an Effectively Artificial Intelligent Algorithm. Micromachines. 2022; 13(8):1208. https://doi.org/10.3390/mi13081208.

[87] Duan, Q; Nguyen V. Q.; Heba, A.; Abdulaziz, A., Optimal scheduling and management of a smart city within the safe framework, IEEE Access, Vol 8 (2020): 161847-161861.

[88] Alsokhiry, F.; Siano, P.; Annuk, A.; Mohamed, M.A. A Novel Time-of-Use Pricing Based Energy Management System for Smart Home Appliances: Cost-Effective Method. Sustainability, 2022, Vol 14, 14556. https://doi.org/10.3390/su142114556

Reviewer 2 Report

Please read the attached file. Thank you.

Author Response

  1. Line 465, 483: Remove the extra space after "."

Answer:   We remove the extra space after "." from Line 465, 483.

  1. Why did you choose the GA and SAA Algorithms (which are old approaches. Genetic algorithms came from the research of John Holland, at the University of Michigan, in 1960 but won't become popular until the 90s.) for this study?

Answer:   Yes, our research apply the concept of Genetics Algorithms from the research of John Holland which still work efficiency until now because the structure not complexity. Therefore, our research team choose the Hybrid Genetic Algorithm and Simulated Annealing Algorithm because it still effective approached for apply to our research to proving the effective recommendation of the tourism systems. However, we put more literature about this algorithms to be more clarify (Line   276-278;  292-308;  443-475;  487-488)

  1. Have you compared its accuracy with that of other approaches? Could you please consider adding these appropriate novel references to your applications for the proposed hybrid algorithms-optimization approaches to enrich your literature review in different fields of study?

(CFD Analysis and Optimum Design for a Centrifugal Pump Using an Effectively Artificial Intelligent Algorithm; Optimizing compliant gripper mechanism design by employing an effective bi-algorithm: fuzzy logic and ANFIS, An efficient hybrid approach of finite element method, artificial neural network-based multiobjective genetic algorithm for computational optimization of a linear compliant mechanism of nanoindentation tester, …)

Answer:   

  • Yes, we evaluate the accuracy of this algorithm by convergence, the plot of cost vs. iterations makes it simple to see how the SAA Algorithm is convergent. Other elements with a similar viewpoint include round-off errors and local extrema (See Figure 6) (Line 644-648; 780-783).
  • We review literature that related to hybrid algorithms-optimization approaches to enrich in different fields of study such as Wang et al. (2022) [69] proposed a cutting-edge strategy to raise the efficiency, power, and head of centrifugal pumps, Duan, et al. (2020) and Alsokhiry et al. (2022), etc. (Line 285-310; 784-775)

  1. What's the best way of calculating the probability that a simulated annealing algorithm will converge to a fixed point?

Answer:   The simulated annealing algorithm will converge to a fixed point in simulated annealing; the acceptance probability for a new state in step k is traditionally defined as

                        Where  is the change in objective function  which is to be                    minimized and  is a strictly decreasing positive sequence with   .

(Line 502-508)

  1. How do you evaluate the accuracy of this research?

Answer:   To evaluate the accuracy of this algorithm by convergence, the plot of cost vs. iterations makes it simple to see how the SAA Algorithm is convergent. Other elements with a similar viewpoint include round-off errors and local extrema. In Figure 6 , the cost function is the minimum beginning at generation 1125 (Line 642-645). These are the most crucial indexes to compare in order to evaluate the algorithm's efficiency and consistency (711-722).

  1. Why do you use the Proposed Hybrid Genetic Algorithm-Simulated Annealing Algorithm, which is quite old, instead of the new Algorithms?

Answer:   Even though the proposed Hybrid Genetic Algorithm-Simulated Annealing Algorithm is a traditional algorithm, it still works effectively for our recommendation system. The hybrid algorithm that is being suggested combines the GA and SA algorithms. The benefits of both algorithms are combined in the suggested framework to get the best answers for the scheduling problems (Line 485-489).

  1. What is the limitation of this study?

Answer:   A limitation of this study is that the method is time consuming, which can cause the amount of services to vary if this method is applied in a large area and result in a very long time for the output result (Line 778-781).

  1. What is the author's following research?

Answer:   The author following the concept of the JSSP can be expressed JOB SHOP SCHEDULING PROBLEM  to solve the problem of this study (Line 443-475).

Reviewer 3 Report

The article shows shortcomings that must be solved. These shortcomings and comments that I propose to incorporate are written as follows:

1.      It is also necessary to supplement and clearly state the goal of the research in the abstract, which is not good presented.

2.      I think the Limitations of the Study in the conclusion part should be in the discussed.

3.      The research and research problem is not clearly supported by current literature.

4.      How the authors determine the parameters for the algorithms.

5.      Correct all the grammatical mistakes and typo errors.

Author Response

  1. It is also necessary to supplement and dearly state the goal of the research in the abstract,   which  is  not  good  presented.

Answer:   We put the aim of this paper in the abstract already (Line  20-25).

  1. 2. I think the Limitations of the Study in the conclusion part should be in the discussed.

Answer:   We put the limitations of the study in the conclusions part (Line  778-781).

  1. 3. The research and research problem is  not clearly supported by current literature.

Answer:   We put the research problem in the introduction section (Line 122-148).

  1. How the authors determine  the parameters  for  the  algorithms.

Answer:   We define the parameter for the algorithms follow the research problem by using  JSSP can be expressed JOB SHOP SCHEDULING PROBLEM.

Each of the "n" jobs consists of a number of procedures that must be carried out on "m" different machines. A single machine is employed for a set amount of time for each operation. Each operation can only be processed by one machine at a time, and processing must continue uninterruptedly until completion once processing on a given machine begins. A specific job's processes must be completed in a specific sequence.

The fundamental goal of the challenge is to plan the activities or operations of the tasks on the machines so that the overall time to complete all of the jobs, or the makespan (Cmax), is as short as possible. The phrase "makespan" describes the total amount of time needed to finish all operations for all tasks on all machines. It is a measurement of the amount of time elapsed between the beginning of the first operation and its conclusion. Sometimes, there may be several solutions with the shortest makespan, but the objective is to identify any one of them; finding all feasible optimum solutions is not always necessary.

(Line 433-484    )

  1. Correct all the grammatical mistakes and typo errors.

Answer:   We already re-check the grammar with the English academic proof center and recheck typing errors.  

Round 2

Reviewer 3 Report

The authors discussed the genetic algorithm, thus I suggested the authors discuss more literatures in this area, For instance, you can refer to doi.org/10.1016/j.jclepro.2019.118845 and doi.org/10.1007/s12541-018-0074-3.

Author Response

We review the two papers following your recommendation in the literature review session in Line 436-442.

  1. Zhang, L.; Zhang, B.; Bao, H.; Huang, H. Optimization of Cutting Parameters for Minimizing Environmental Impact: Considering Energy Efficiency, Noise Emission and Economic Dimension, International Journal of Precision Engineering and Manufacturing,2018, Vol. 19, No. 4, pp. 613-624, DOI: 10.1007/s12541-018-0074-3. (Line 987-989)
  2. Zhang, B; Xu, L; Zhang,J. A multi-objective cellular genetic algorithm for energy-oriented balancing and sequencing problem of mixed-model assembly line, Journal of Cleaner Production, 2020, Vol. 244, 2020, 118845, doi: https://doi.org/10.1016/j.jclepro.2019.118845. (Line 990-992)